# Does Orthodontic Treatment Change the Preferred Chewing Side of Patients with Malocclusion?

**DOI:** 10.3390/jcm11216343

**Published:** 2022-10-27

**Authors:** Shuko Arai, Chiho Kato, Ippei Watari, Takashi Ono

**Affiliations:** Department of Orthodontic Science, Graduate School of Medical and Dental Sciences, Tokyo Medical and Dental University (TMDU), Tokyo 113-8519, Japan

**Keywords:** lateral mandibular movement, laterality, mandibular kinesiograph, mastication, occlusal canting, orthodontic treatment, orthodontics, preferred chewing side

## Abstract

Whether orthodontic treatment can change the preferred chewing side (PCS) is unknown. This study examined (1) if the PCS changes after orthodontic treatment and (2) which factors contribute to this change. Two hundred fifty patients who visited the orthodontic clinic at Tokyo Medical and Dental University Hospital between 2017 and 2020 were included in the study. Mandibular kinesiograph (MKG) was taken at pre- and post-treatment, and PCS was determined. Patients who showed a change in PCS to the opposite side and those who showed no change in PCS at post-treatment were pooled into the PCS-changed and PCS-unchanged groups, respectively. The demographic, clinical, and cephalometric parameters were compared between the groups. Significant factors associated with changes in were of age < 20 years at the beginning of orthodontic treatment (odds ratio (OR), 2.00), maximum lateral mandibular movement to PCS ≥ 10.0 mm at pre-treatment (OR, 6.51), and change in occlusal canting of ≥1.0° (OR, 2.72). The predicted probability of change in PCS was 13.2%, 36.0%, and 67.5% for no factor, one factor, and two factors associated with PCS change, respectively. Orthodontic treatment may change PCS due to patient age, maximum lateral mandibular movement to PCS, and change in occlusal canting.

## 1. Introduction

Although organs are nearly symmetrical in appearance, such as eyes and hands, they often have functional differences between the left and the right side [1]. The side of the body that is used most often is considered to be the dominant side of the organ (i.e., sidedness), and the organs that have sidedness are the hands, feet, eyes, and ears [2]. Studies have been conducted from the perspective of genetic factors [3,4] given that sidedness is determined by the brain [1,5]. Moreover, the sidedness of organs that we are aware of and what was determined by the brain often coincide [6]. Considering this, the dominant arm may be replaced due to changes in brain function when it is amputated due to trauma [7] and brain damage [8]. These findings indicate that the brain is a major determinant of sidedness.

There is also sidedness in the masticatory function, referred to as the habitual chewing side or preferred chewing side (PCS) [9,10], which is observed in both children and adults [10,11]. PCS is more prevalent on the right than the left side in adults [12,13,14]. There is a relationship between PCS and the sidedness of various organs determined by the brain (e.g., dominant hand, foot, eye, and ear) [1,5], with some reports suggesting a strong relationship with the dominant ear [15,16], while others indicate that there is no association [17]. Moreover, a significant relationship between PCS, tongue movement, and the concomitant activation of the primary sensorimotor cortex (S1/M1) has been demonstrated by functional magnetic resonance imaging (fMRI), suggesting that the activation of the S1/M1 was significantly greater on the side contralateral to PCS [18,19]. Meanwhile, acquired factors such as pain due to temporomandibular joint (TMJ) disorder and asymmetrical tooth loss after extraction may cause differences between the left and right sides of occlusion and the masticatory system [20]. Thus, changes in the oral environment may alter PCS. 

Previous studies have suggested that having PCS is mostly due to left-right differences in occlusal force and masticatory efficiency [15,17] and does not affect the jaw and TMJ function [21]; however, it is still controversial. Orthodontic treatment not only changes the teeth alignment but also improves occlusal conditions and facilitates masticatory function in terms of jaw movement and masticatory muscle activity [22,23]. For example, it can change the masticatory path of a patient with unilateral mastication due to crossbite from reverse to a normal pattern to achieve bilateral mastication [24,25]. Moreover, the masticatory function can be improved in orthodontic treatment by changing and improving the anteroposterior relationship of the maxillofacial skeleton, an inclination of the occlusal plane, and deviation of the mandible [26].

However, it is unknown whether the peripheral change by orthodontic treatment can change PCS. The possibility of orthodontic treatment-induced changes in the PCS suggests the plasticity of the masticatory function associated with orthodontic treatment [20]. In addition, determining the environmental factors associated with the change in PCS may lead to the acquisition of a closer and functionally superior occlusal relationship and masticatory function. Therefore, this study examined [1] if PCS changes after orthodontic treatment, and [2] which factors contribute to this change and hypothesized that no physical or clinical factors are associated with PCS changes in orthodontic treatment.

## 2. Materials and Methods

### 2.1. Patients 

A total of 3050 patients with malocclusion, who visited the orthodontic clinic at Tokyo Medical and Dental University Hospital between 2017 and 2020, were initially selected for this retrospective study. Either the patients or the patients’ guardians were informed and signed an informed consent form approved by the TMDU Research Ethics Committee (Code: D2018-033). All experiments were carried out in accordance with the principles of the Helsinki Declaration. The inclusion criteria were: (1) age ≥ 16 years before orthodontic treatment, and (2) jaw movement examined using a mandibular kinesiograph (MKG) pre- and post-treatment. The exclusion criteria were: (1) previous orthodontic treatment, (2) orofacial pain, TMJ sound, joint pain, jaw movement disorders, and past medical history, and (3) patients with residual post-operative numbness that interferes with jaw movement. After applying the inclusion and exclusion criteria, 250 patients were finally included in the study out of 3050 patients with malocclusion. The number of left and right chewing strokes for 20 strokes were counted using MKG during free gum chewing to determine PCS. If there was a side with ≥ 70% number of strokes, the patient was defined as exhibiting PCS. MKG recording was repeated after active orthodontic treatment. The patients who changed PCS from one side to the other and those who had the same PCS after orthodontic treatment were included for further analysis to investigate the contributing factors, including skeletal and dentoalveolar parameters. Those who had no PCS after orthodontic treatment were excluded for further analysis. The data presented in this study are available on request from the corresponding author. The data are not publicly available due to the risk of personal information leakage.

### 2.2. Variables for Measurement

Information regarding sex, age, and tooth extraction was obtained from clinical records. Canine and molar anteroposterior relationships, open bite, and crossbite were recorded from the dental cast. All patients have an ideal molar relationship after completion of orthodontic treatment: no categorization of malocclusion was necessary. The patient’s head was affixed with the ear rods, and the Frankfurt plane was made parallel to the floor when postero-anterior and lateral cephalometric radiographs (Hyper-X, ASAHI., Tokyo, Japan) were taken [27,28,29]. The source-to-subject and subject-to-film distances were always fixed. The following parameters were determined: ANB angle, mandibular plane angle, mandibular shift [30,31], and occlusal canting [32]. The amount of mandibular shift was measured through the distance of the skeletal sagittal midline to the mental spine. The facial sagittal midline is constructed from the crista galli through the anterior nasal spine (ANS) to the chin area (Figure 1). Occlusal canting is represented by the angle of the maxillary intermolar (M1-M1) plane and the line that is perpendicular to the skeletal sagittal midline. The aforementioned values were measured three times on different days seven days apart, using software (Winceph ver.9.0, Rise Corp., Tokyo, Japan), and the average value was calculated. The cephalometric radiographs were traced and analyzed by a single investigator (SA). Statistical analyses were performed to determine possible correlations among groups. All parameters of cephalometric radiographs were randomly re-measured, and errors were calculated by Dahlberg’s formula [33]; on average, the method error was 0.15° for the ANB angle, 0.05 mm for the mandibular shift and 0.01° for the occlusal canting. Inter-group comparisons were carried out using a one-way analysis of variance and Chi-square test.

Jaw movements were recorded using a mandibular kinesiograph (MKG) (K-7 Evaluation System; JM Ortho Inc., Tokyo, Japan), which traces the mandibular movement in three dimensions of a guide magnet. The guide magnet was fixed to the mandibular incisors ruling out contact with the maxillary incisors. The system does not constrain mandibular movement in any way and comprises (a) the guide magnet, (b) a sensor array, and (c) the display and analysis system [34,35]. The patients were seated on a chair with their backs straight without head support, and with the arms resting on the legs and both eyes open. The procedure was performed by a single operator (SA) in a quiet room. The sensor array was affixed to the patient’s head, and the Frankfurt plane was made parallel to the floor when mandibular movement factors (jaw movement type and maximum lateral mandibular movement to PCS) were taken. The maximum lateral mandibular movement to PCS during free gum chewing was measured through the distance perpendicular to the Frankfurt plane from the closed position.

### 2.3. Outcome

The number of left and right strokes was counted during free hard gum chewing using MKG to determine PCS. Chewing gum was placed in the middle of the patient’s tongue, and they were instructed to chew freely, while the masticatory movement was recorded using MKG. Patients were not informed that chewing laterality was the item of interest to avoid bias and awareness of the chewing side. The number of left and right chewing strokes for 20 strokes from the fourth stroke was counted except for the first three strokes, and when there was a side with ≥70% number of strokes, the side was determined as PCS (PCS (R) = number of right stroke/20 ≥ 0.7, PCS (L) = number of left stroke/20 ≥ 0.7) [17,36]. It was determined that there was no PCS when there was no preferred side of ≥ 70% [17,36]. This test was carried out during pre- and post-treatment, and PCS was examined. Patients who changed “left to right” and “right to left” after orthodontic treatment were pooled into the “PCS-changed (cPCS)” group. In addition, patients who exhibited PCS before orthodontic treatment and had the same PCS after orthodontic treatment were pooled into the “PCS-unchanged (uPCS)” group.

### 2.4. Statistical Analyses

Patient characteristics were compared between the cPCS and uPCS groups. The factors associated with the change in PCS were explored. Logistic regression analysis was used to estimate the odds ratios (ORs). Assuming a type 1 error of 0.05 and a power of 0.8, 80 subjects (30 for the intervention group and 50 for the control group) would be needed to detect a statistical significance for an odds ratio of 2.0. Univariate logistic regression models have used wherein the change in PCS was regressed on differences in the demographic (i.e., age and sex), clinical (i.e., orthognathic surgery or not, and maxillary or mandibular premolar or molar extraction), cephalometric (i.e., ANB angle, mandibular plane angle, mandibular shift, and occlusal canting), dental cast (i.e., canine and molar occlusal relationships were symmetrical or asymmetrical, anterior and molar open bite or not, and anterior and molar crossbite or not), and MKG (i.e., jaw movement type and maximum lateral mandibular movement to PCS during free gum chewing) parameters. Several continuous variables were classified into categorical groups based on clinical relevance. Age was classified as <20 years and ≥20 years as previously reported [37]. The cut-off for the change in occlusal canting was determined as previously reported, wherein a change by ≥1° is mainly due to surgical orthodontic treatment and orthodontic treatment with mini-implants or miniplates [34,35,37]. The maximum lateral mandibular movement to PCS during free gum chewing was classified as <10 mm and ≥10 mm as previously reported [38,39]. Subsequently, multivariable logistic regression models were used including (1) age, (2) change in occlusal canting, and (3) maximum lateral mandibular movement to PCS during free gum chewing. A model predicting change in PCS using three variables included in the multivariable logistic regression models was also developed. For clinical use, one point was assigned for each factor associated with the change in PCS (i.e., 0–3 points). Afterward, they were categorized into three groups to obtain reliable estimates: 0 point, 1 point, or 2 points [40]. The observed probability was calculated for the three groups and then compared to the predicted probability based on the multivariable logistic regression model. All analyses were performed using the software, Stata version 15 (Stata Corp, College Station, TX, USA).

## 3. Results

### 3.1. Baseline Characteristics

Before orthodontic treatment, 121 patients (48.4%, 121/250) exhibited PCS; 69 patients preferred the right side, while 52 patients preferred the left side. Meanwhile, 129 patients (51.6%, 129/250) had no PCS. Among those who exhibited PCS at pre-treatment, 108 patients (89.3%, 108/121) showed PCS, while 13 patients (10.7%, 13/121) did not show PCS after orthodontic treatment. In the cPCS group, 17 patients (15.7%, 17/108) changed PCS “from right to left”, while 16 patients (14.8%, 16/108) changed “from left to right”. Meanwhile, in the uPCS group, the right PCS remained in 39 patients (36.1%, 39/108), while the left PCS remained in 36 patients (33.3%, 36/108).

A total of 108 patients who showed PCS at pre- and post-treatment were further analyzed. Table 1 shows their baseline characteristics. They are aged between 16 and 53 years, with a mean of 23.3 years (<20 years, 43 (39.8%); ≥20 years, 65 (60.2%)). Among the participants, 54 (50.0%) underwent general orthodontic treatment, 54 (50.0%) received surgical orthodontic treatment, 55 (50.9%) had a maxillary premolar or molar extraction, and 28 (25.9%) had mandibular premolar or molar extraction.

### 3.2. Univariate Logistic Regression Analysis

Table 2 shows the ORs for predicting factors for PCS changes in the univariate logistic regression analysis. Patient characteristics were compared between the cPCS and uPCS groups. Age < 20 years was significantly associated with PCS change as compared to age ≥ 20 (OR: 2.00, 95% confidence interval (CI): 0.87–4.60) and change in occlusal canting ≥ 1.0° was significantly associated with the change in PCS compared to the change in occlusal canting < 1.0° (OR: 2.01, 95% CI: 1.08–3.73). In addition, maximum lateral mandibular movement to PCS during free gum chewing ≥ 10.0 mm was significantly associated with the change in PCS compared to the maximum lateral mandibular movement to PCS < 10 mm (OR: 6.51, 95% CI: 1.19–35.6).

### 3.3. Multivariate Logistic Regression Analysis

Clinical, demographic, cephalometric, dental cast, and MKG parameters were included in the multivariable logistic regression analysis.

### 3.4. A model to Predict Changes in PCS

Finally, a model predicting changes in PCS based on the number of factors associated with PCS change included in the multivariable logistic regression model was developed (Table 3).

The number of factors associated with the change in PCS was assigned as two when there were two or more factors. In this model, one point was assigned for each of the three factors associated with PCS change such that the factor associated with the change in PCS had a score ranging from 0 (i.e., no factor) to 3 points (i.e., all factors). Table 3 shows the predicted and observed probabilities of PCS change. Overall, 41 patients had no factor (i.e., 0 points), 56 had 1 factor, and 11 had two or more factors. The proportion of those for whom PCS change was observed was 17.1% (7/41) for no factor associated with the change in PCS, 30.4% (17/56) for one factor, and 81.8% (9/11) for two factors. These proportions were well predicted in the multivariable logistic regression model: the predicted probability of PCS change was 13.2% for no factor, 36.0% for one factor, and 67.5% for two factors (Figure 2).

## 4. Discussion

This study suggested that orthodontic treatment might change PCS owing to age, change in occlusal canting, and maximum lateral mandibular movement to PCS. A model predicting changes in PCS based on the number of factors associated with the change in PCS was developed. Moreover, this prediction model can identify patients with factors associated with changes in PCS.

Masticatory movement is a rhythmic activity dominated by the cerebral hemispheric organization (pattern generator) and modified by peripheral stimulation. Therefore, the feedback of information from the periphery regulates masticatory patterns [41]. Physiologically, the masticatory function is observed on both sides; however, PCS is mainly used [9]. In patients with right PCS with bilateral tooth clenching, the left S1M1 was more active than the right side [42], suggesting that there is a connection between the brain and the oral function. In this study, 121 (48.4%, 121/250) patients exhibited PCS at the beginning of the orthodontic treatment. Meanwhile, PCS has been found in 45.0% [14,20,36], 56.0% [15], 63.3% [17], 76.0% [16], 78.0% [9], and 97.9% [43] of the patients in previous studies. The discrepancy in the literature regarding PCS incidence may be due to differences in the sample population and the definition of PCS. In this study, PCS was defined as ≥70% of either the left or right stroke during free gum chewing. Patients who had a difference between the left and right masticatory strokes but did not meet this value were excluded, resulting in a lower incidence of PCS than in previous studies. In this study, 69 patients (27.6%) had right PCS, while 52 (20.8%) had left PCS. The incidence of PCS on the left and right sides was similar to previous studies reporting that: 30% had the right side and 15% had the left [15], 40.0% had the right side and 36% had the left [17], and 33% had the right side and 11.9% had the left [43] in the permanent dentition.

In this study, the average duration of orthodontic treatment was 34 months. PCS, the dominant side (laterality) of mastication, is principally determined by brain dominance, such as the dominant hand, foot, eye, and ear [1,5]. Handedness reflects the inherent specialization in the cerebral hemispheric control processes (dominant system for controlling limb trajectory and non-dominant system for controlling limb position) [44]. If a left-handed child is trained to write with the right hand from childhood, activities in the cerebral sensorimotor areas are invariant [45]. Thus, handedness is essentially unchanged throughout life. Meanwhile, when a disability such as a hand amputation occurs, the equilibrium between the left and right hemispheres is reorganized, and the lateralization of the brain changes at the early stage of the amputation injury [7]. Moreover, bilateral gum chewing contributes to a temporary increase in activation of the brain ipsilateral to PCS in tongue movements; brain activation is still significantly greater on the opposite side of the PCS [46]. For PCS, no studies with natural time courses have been reported. However, as with the dominant hand, brain dominance is involved in determining the dominant side. Moreover, brain changes do not occur over time and changes in the dominant side do not occur naturally [1,3,7]. However, if the appearance of malocclusion (peripheral changes) had altered the PCS, which is originally determined by the brain, it is possible that orthodontic treatment could restore the original laterality; this was the subject of investigation in this study.

In this study, PCS was more likely to change after orthodontic treatment in those <20 years of age than in those ≥20 years of age. Growth in muscle, bone, and height ends at ages 20–22 in males and 18–20 in females [37]. In the oral cavity, the maximum occlusal force increases until age 20 and remains stable until age 50 [47]. From the above, growth is completed by the age of 20 years, and the masticatory environment is completed. PCS has a larger occlusal force and occlusal contact area than non-PCS [17]. Thus, if the strength of the left and right occlusal forces switches during the growth process, it may lead to changes in PCS. Orthodontic treatment for young people can help them gain proper masticatory ability [22] and send appropriate feedback to the brain. It is inferred that orthodontic treatment at the age of <20 years before the masticatory environment becomes stable, brings appropriate function, which normalizes the central output and leads to changes in the PCS.

In this study, the change in occlusal canting ranged from 0° to 3.0°, with a mean change of 0.36°. Patients with a change in occlusal canting of ≥1° before and after orthodontic treatment showed more changes in PCS after orthodontic treatment. More changes in occlusal canting are observed in patients undergoing orthognathic treatment than in patients undergoing general orthodontic treatment because orthodontic treatment involving surgery can greatly change the occlusal canting [48].

Recently, the application of the mini-screw/mini-plate produced changes in occlusal canting through orthodontic forces under pressure to the unilateral molars [49,50]. Changes in occlusal canting were observed in some cases even in orthodontic treatment only. Occlusal canting is an indicator of facial symmetry [51,52]. Change in occlusal canting may affect facial and masticatory muscle activity, resulting in changes in the left-right occlusal force and left-right facial height. Moreover, occlusal canting is often accompanied by mandibular deviation [48,49]. Mandibular deviation causes an alteration in the afferent information by altering the response properties of the TMJ mechanoreceptors [53]. Orthodontic treatment can result in symmetrical function [22]. It is possible that the change in occlusal canting through orthodontic treatment altered the occlusion (peripherally), and the change in afferent information affected the central (brain) output, which in turn altered PCS. Meanwhile, anteroposterior and vertical changes due to orthodontic treatment did not affect the change in PCS. Changes in occlusal vertical dimension induced cortical plasticity [54] and influenced the response properties of the masticatory muscle [55]; however, PCS was not affected. Consequently, PCS may be more responsive to changes in the left-right input to the brain.

MKG analysis revealed that the maximum lateral mandibular movement to PCS during free gum chewing before orthodontic treatment ranged from 2.9 mm to 13.5 mm (mean: 5.7 mm); however, PCS was often altered when patients with a lateral motion limit of ≥10 mm received orthodontic treatment. Since the maximum lateral mandibular movement of healthy patients is within 10 mm [38,39], a marginal motion of ≥10 mm during free gum chewing at the start of the orthodontic treatment is expected to indicate unstable jaw movement. 

Although mandibular movement is defined by the form of the mandibular fossa and condyle, the supero-posterior, posterior, and outer joint spaces on the PCS are smaller than those on the non-PCS in the relationship between PCS and TMJ [56]. Patients whose maximum lateral mandibular movement was larger on the PCS may have a larger joint space on the PCS. The true PCS might have been the opposite side, which might have changed back to the true PCS because orthodontic treatment improved the oral environment. More detailed studies of TMJ morphology, mandibular joint cavity, and mandibular fossa will be required for patients whose PCS changed after orthodontic treatment.

Finally, a model was developed to identify patients whose PCS changed after orthodontic treatment by incorporating three variables: age, change in occlusal canting, and maximum lateral mandibular movement to PCS during free gum chewing. The model showed that patients with 0, 1, and 2 risk factors showed an estimated 17.1%, 30.4%, and 81.8% probability of change in PCS, respectively. Two parameters (age and maximum lateral mandibular movement to PCS during free gum chewing) were evaluated before orthodontic treatment and change in occlusal canting was calculated as a moving predicted value before orthodontic treatment. Orthodontists can use this model for treatment planning or patient counselling. It may help in discussing PCS-conscious treatment and maintenance plans before the procedure. Gaining original PCS can allow for smoother central and peripheral function, which can contribute to not only a closer and more appropriate occlusal relationship through orthodontic treatment but also functional occlusion. 

This study analyzed patients who exhibited PCS before treatment. However, patients who exhibited PCS after but not before treatment and those who exhibited PCS before but disappeared after treatment were not examined. Investigations including such patients may be necessary, and clarification of the relationship between changes in PCS and brain function should be the subject of further studies.

## 5. Conclusions

The findings of this study indicate that PCS may change after treatment owing to three factors: age of <20 years, having a maximum lateral mandibular movement to PCS during free chewing of ≥10 mm at the start of the orthodontic treatment and expecting a change in the occlusal canting of ≥1° due to orthodontic treatment. In addition, the use of a model predicting changes in PCS in orthodontic treatment would be expected to achieve smooth masticatory function.

## Figures and Tables

**Figure 1 jcm-11-06343-f001:**
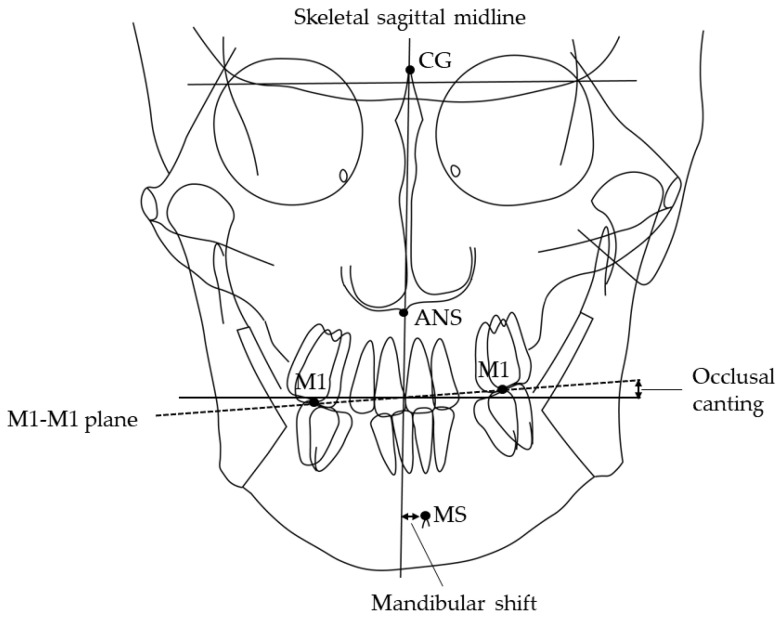
Landmarks and planes used in the cephalometric analysis. MS, mental spine; CG crista galli; ANS, anterior nasal spine. Skeletal sagittal midline, the plane from CG through the ANS; M1-M1 plane, the maxillary intermolar plane; Shift, deviation of MS to the skeletal sagittal midline in the postero-anterior cephalometric radiograph. Occlusal canting, the angle of the M1-M1 plane and the line perpendicular to the skeletal sagittal midline in the postero-anterior cephalometric radiograph.

**Figure 2 jcm-11-06343-f002:**
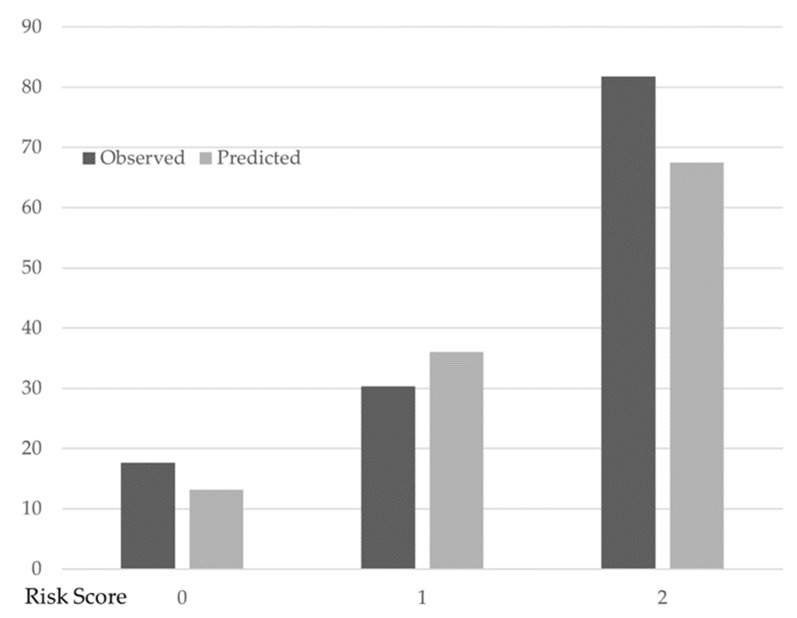
Observed and predicted the probability of PCS change with different factors associated with the change in PCS scores. Filled bars indicate the observed probability of the change in PCS. Open bars indicate the predicted probability of the change in PCS based on multivariable logistic regression analysis. Change in PCS: 7 (factor score: 0), 17 (factor score: 1), 2 (factor score: 2). No change in PCS: 34 (factor score: 0), 39 (factor score: 1), 9 (factor score: 2).

**Table 1 jcm-11-06343-t001:** Patient characteristics.

Factors	Total (*n* = 108)
Demographic	
Age (years, mean ± SD)	23.3 ± 8.1
<20 (%)	43 (39.8)
≥20 (%)	65 (60.2)
Sex	
Male (%)	36 (33.3)
Female (%)	72 (66.7)
Clinical	
With orthognathic surgery	
Yes (%)	54 (50.0)
No (%)	54 (50.0)
With maxillary premolar or molar extraction	
Yes (%)	55 (50.9)
No (%)	53 (49.1)
With mandibular premolar or molar extraction	
Yes (%)	28 (25.9)
No (%)	80 (74.1)
Cephalometric	
Change in ANB angle (degree, mean ± SD)	1.8 ± 3.8
Change in mandibular plane angle (degree, mean ± SD)	1.8 ± 3.4
Change in mandibular plane angle (mm, mean ± SD)	0.8 ± 1.3
Change in mandibular plane angle	0.4 ± 6.7
<1 (%)	78 (72.2)
≥1 (%)	30 (27.8)
Dental	
Molar relationship	
Symmetry (%)	89 (82.4)
Asymmetry (%)	19 (17.6)
Canine relationship	
Symmetry (%)	84 (77.8)
Asymmetry (%)	24 (22.2)
Anterior cross bite	
Yes (%)	55 (57.8)
No (%)	51 (47.2)
Anterior open bite	
Yes (%)	89 (82.4)
No (%)	19 (17.6)
Posterior cross bite	
Yes (%)	36 (33.3)
No (%)	72 (67.7)
Posterior open bite	
Yes (%)	44 (59.3)
No (%)	64 (40.7)
Mandibular kinesiograph	
Jaw movement	
chopping type (%)	30 (27.8)
grinding type (%)	78 (72.2)
Maximum lateral movement to PCS (mm, mean ± SD)	5.7 ± 2.9
<10 (%)	101 (93.5)
≥10 (%)	7 (6.5)

PCS, preferred chewing side; SD, standard deviation.

**Table 2 jcm-11-06343-t002:** Univariate logistic regression analysis in the PCS-unchanged (uPCS) and PCS-changed (cPCS) groups.

Exposures	uPCS (*n* = 75)	cPCS (*n* = 33)	Univariate Logistic Regression		
			OR	95% CI	*p* Value
Demographic					
Age					
<20 (%)	26 (34.7)	17 (51.5)	2.00	0.87–4.60	0.10
≥20 (%)	49 (65.3)	16 (48.5)	1 (reference)		
Sex					
Male (%)	21 (28.0)	15 (45.5)	2.14	0.91–5.02	0.08
Female (%)	54 (72.0)	18 (54.5)	1 (reference)		
Clinical					
With orthognathic surgery					
Yes (%)	38 (50.6)	16 (49.5)	0.92	0.40–2.07	0.84
No (%)	37 (49.3)	17 (51.5)	1 (reference)		
With maxillary premolar or molar extraction					
Yes (%)	39 (52.0)	16 (48.5)	0.87	0.38–1.97	0.74
No (%)	36 (48.0)	17 (49.5)	1 (reference)		
With mandibular premolar or molar extraction					
Yes (%)	22 (29.3)	6 (18.2)	0.53	0.19–1.47	0.23
No (%)	53 (70.6)	27 (81.8)	1 (reference)		
Cephalometric					
Change in ANB angle (degree, mean ± SD)	1.69 ± 2.87	2.13 ± 5.35	1.03	0.98–1.14	0.57
Change in mandibular plane angle (degree, mean ± SD)	1.99 ± 3.60	1.48 ± 3.04	0.95	0.84–1.08	0.48
Change in mandibular plane angle (mm, mean ± SD)	0.60 ± 1.00	1.15 ± 1.67	1.39	1.01–1.92	0.05
Change in mandibular plane angle (degree, mean ± SD)	0.27 ± 0.55	0.57 ± 0.79	2.01	1.08–3.73	0.03
<1 (%)	59 (78.7)	19 (57.6)	1 (reference)		
≥1 (%)	16 (21.3)	14 (42.4)	2.72	1.12–6.57	0.03
Dental					
Molar relationship					
Symmetry (%)	64 (85.3)	23 (75.8)	1 (reference)		
Asymmetry (%)	11 (14.7)	8 (24.2)	1.86	0.67–5.17	0.23
Caine relationship					
Symmetry (%)	60 (80.0)	24 (72.7)	1 (reference)		
Asymmetry (%)	15 (20.0)	9 (27.3)	1.50	0.58–3.89	0.40
Anterior cross bite					
Yes (%)	38 (50.7)	19 (57.6)	1.32	0.58–3.02	0.50
No (%)	37 (49.3)	14 (42.4)	1 (reference)		
Anterior open bite					
Yes (%)	64 (85.3)	25 (75.8)	0.54	0.19–1.49	0.23
No (%)	11 (14.7)	8 (24.4)	1 (reference)		
Posterior cross bite					
Yes (%)	23 (30.7)	13 (39.4)	1.47	0.63–3.45	0.38
No (%)	52 (69.3)	20 (60.6)	1 (reference)		
Posterior open bite					
Yes (%)	27 (36.0)	17 (51.5)	1.89	0.82–4.33	0.13
No (%)	48 (64.0)	16 (48.5)	1 (reference)		
Mandibular kinesiograph					
Jaw movement					
chopping type (%)	19 (25.3)	11 (33.3)	1.47	0.60–3.59	0.39
gridding type (%)	56 (74.7)	22 (66.7)	1 (reference)		
Maximum lateral movement to PCS					
<10 (%)	73 (97.3)	28 (84.9)	1 (reference)		
≥10 (%)	2 (2.7)	5 (15.1)	6.51	1.19–35.56	0.03

CI, confidence interval; cPCS, PCS-changed; OR, odds ratio; PCS, preferred chewing side; SD, standard deviation; uPCS, PCS-unchanged.

**Table 3 jcm-11-06343-t003:** Multivariable logistic regression analysis for clinical factors associated with PCS change.

	Multivariate Analysis			Assigned Point
Variables	OR	95% CI	*p* Value	
Age (years, ref ≥ 20)				
<20	3.23	1.23–8.45	0.02	=1 if age < 20
Occlusal canting change (degree, ref < 1)				
≥1	3.55	1.29–9.84	0.01	=1 if angle ≥ 1
Maximum mandibular lateral movement to PCS(mm, ref < 10)				
≥10	5.76	0.99–33.4	0.05	=1 if lateral movement to PCS ≥ 10

Total possible score: 0 (no factor) to 3 (all factors). CI, confidence interval; OR, odds ratio; PCS, preferred chewing side.

## Data Availability

The data underlying this article are available upon reasonable request to the corresponding author.

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
