# Peer review of "Does Orthodontic Treatment Change the Preferred Chewing Side of Patients with Malocclusion?"

_jcm, 2022, doi:10.3390/jcm11216343_

Round 1

Reviewer 1 Report

The goal of this clinical study is to examine  if the preferred chewing side changes after orthodontic treatment, and which factors contribute to this change.

This knowledge produced is relevant to the field of the Journal. This research is under the scope of this Journal.

There are some aspects which is possibly improved in the various sections of the manuscript:

  • Correct typos in all manuscript. 
  • The use of personal pronouns should be avoided. Example“We examined…”

(Abstract)

-  In the results, is important to show more information, add some of the values.

(Keywords)  

  • Please more keywords, and order these keywords / Mesh Terms alphabetically for a standardized presentation of the keywords.

(Introduction)

  • Identified the aim and  null hypothesis on the end of the introdution.

(Results)

  • Improve the resolution quality of all figures and graphs (and a presentation). The font/language in the figure/caption is different from the text. Please, standardize the size and the font in the figures with the font of the manuscript. 

(Discussion)

- In  clarified others limitations of this study?And, add more the future perspectives. Please, change the title the point 5 “study limitations” to “study limitations a future perspectives” or inclued in discussion. 

(References)

  • References are not standardized. The titles of references have a different format, the title of the article is written in capital letters at the beginning of words, others only in lower case.
  • Please carefully review all references to ensure their accuracy.

Reviewer 2 Report

It is a very interesting research, but the manuscript needs to be corrected.

11.      Line 82 to 86 – technical information are included into the manuscript.

22.      Line 93 – in that place it should be pointed that PA cephalometric (CEF) radiographs was taken. It is important, because usually lateral CEF are analysed for orthodontic diagnostic and treatment planning.

33.      Line 93 – the equipment used for CEF examination should be described.

44.      Line 95 – if you analysed ANB angle the lateral CEF was necessary. So two different x-rays was taken for the research. It need to be added to MM.

55.      Line 96 and so – past tens should be used to describe examination methods.

66.      Line 127 – past tens

77.      Line 172 – fig. 2 – it is no explanation in any part of the manuscript about that chart.

88.      There was no any information in MM about orthodontic treatment analysis except tooth extractions, while in the result part all carried out treatment was analysed in details. It need to be changed (in MM or in results part)

99.      From 197 line to 207 – it is duplication of the data from the table. That could be avoided from the text.

110.   All data from the Table 1 are presented also in Table 2, so the Table 2 would be enough for the presenting of research results.

111.   The sentence started in line 255 about founding of PCS by different researchers is too concise.

112.   Line 262 – in all reported articles the right side was more often the PCS. Did you find any connection between the PCS and the method of infant feedings? Some authors (prof. R. Slavicek among others) suggests that breast feeding protects side discrepancy due to necessity of right and left side feeding in opposite to handling the bottle by right handed parents and prefer one side during feeding.

113.   Conclusions of the study need to be changed. In present form it is the summary of results.

114.   The reference numbering is duplicated.

Round 2

Reviewer 2 Report

I am satisfied with your correction.